# Combined Salivary Proteome Profiling and Machine Learning Analysis Provides Insight into Molecular Signature for Autoimmune Liver Diseases Classification

**DOI:** 10.3390/ijms241512207

**Published:** 2023-07-30

**Authors:** Giulia Guadalupi, Cristina Contini, Federica Iavarone, Massimo Castagnola, Irene Messana, Gavino Faa, Simona Onali, Luchino Chessa, Rui Vitorino, Francisco Amado, Giacomo Diaz, Barbara Manconi, Tiziana Cabras, Alessandra Olianas

**Affiliations:** 1Dipartimento di Scienze della Vita e dell’Ambiente, Università di Cagliari, 09124 Cagliari, Italy; giulia.guadalupi@unica.it (G.G.); cristina.contini93@unica.it (C.C.); tcabras@unica.it (T.C.); olianas@unica.it (A.O.); 2Fondazione Policlinico Universitario IRCCS “A. Gemelli”, 00168 Rome, Italy; federica.iavarone@unicatt.it; 3Dipartimento di Scienze Biotecnologiche di Base, Cliniche Intensivologiche e Perioperatorie, Università Cattolica del Sacro Cuore, 00168 Rome, Italy; 4Laboratorio di Proteomica, Centro Europeo di Ricerca sul Cervello, IRCCS Fondazione Santa Lucia, 00168 Rome, Italy; maxcastagnola@outlook.it; 5Istituto di Scienze e Tecnologie Chimiche “Giulio Natta”, Consiglio Nazionale delle Ricerche, 00168 Rome, Italy; imessana53@gmail.com; 6Division of Pathology, Department of Medical Sciences and Public Health, University Hospital, 09124 Cagliari, Italy; gavinofaa@gmail.com; 7Liver Unit, University Hospital of Cagliari, 09124 Cagliari, Italy; simona.onali@yahoo.it (S.O.); luchinochessa@unica.it (L.C.); 8iBiMED, Department of Medical Science, University of Aveiro, 3810-193 Aveiro, Portugal; rvitorino@ua.pt; 9UnIC@RISE, Department of Surgery and Physiology, Faculty of Medicine of the University of Porto, 4200-319 Porto, Portugal; 10LAQV/REQUIMTE, Department of Chemistry, University of Aveiro, 3810-193 Aveiro, Portugal; famado@ua.pt; 11Dipartimento di Scienze Biomediche, Università di Cagliari, 09124 Cagliari, Italy; gdiaz@unica.it

**Keywords:** autoimmune hepatitis, autoimmune liver diseases, bottom-up proteomics, Cofilin-1, mass spectrometry, hornerin, primary biliary cholangitis, saliva, random forest

## Abstract

Autoimmune hepatitis (AIH) and primary biliary cholangitis (PBC) are autoimmune liver diseases that target the liver and have a wide spectrum of presentation. A global overview of quantitative variations on the salivary proteome in presence of these two pathologies is investigated in this study. The acid-insoluble salivary fraction of AIH and PBC patients, and healthy controls (HCs), was analyzed using a gel-based bottom-up proteomic approach combined with a robust machine learning statistical analysis of the dataset. The abundance of Arginase, Junction plakoglobin, Desmoplakin, Hexokinase-3 and Desmocollin-1 decreased, while that of BPI fold-containing family A member 2 increased in AIHp compared to HCs; the abundance of Gelsolin, CD14, Tumor-associated calcium signal transducer 2, Clusterin, Heterogeneous nuclear ribonucleoproteins A2/B1, Cofilin-1 and BPI fold-containing family B member 2 increased in PBCp compared to HCs. The abundance of Hornerin decreased in both AIHp and PBCp with respect to HCs and provided an area under the ROC curve of 0.939. Machine learning analysis confirmed the feasibility of the salivary proteome to discriminate groups of subjects based on AIH or PBC occurrence as previously suggested by our group. The topology-based functional enrichment analysis performed on these potential salivary biomarkers highlights an enrichment of terms mostly related to the immune system, but also with a strong involvement in liver fibrosis process and with antimicrobial activity.

## 1. Introduction

Autoimmune hepatitis (AIH) and primary biliary cholangitis (PBC) are two types of autoimmune liver diseases (AILDs) that target the liver and are characterized by a wide spectrum of presentation.

AIH refers to chronic and progressive inflammation of the liver from an unknown cause but is thought to be a combination of genetic predisposition, environmental triggers and failure of the native immune system, which results in chronic inflammation of the liver, necrosis of hepatocyte and subsequent fibrosis. AIH is a rare worldwide disease predominant in women by a ratio of 4:1 with respect to men [1,2]. There are two known types of AIH: the most common Type 1 involves anti-smooth muscle antibodies (ASMA) with or without anti-nuclear antibodies (ANA). The more rare and often more severe Type 2 tends to appear earlier, usually during childhood, and progresses faster than type 1; it involves anti-liver/anti-kidney microsome (anti-LMK) type 1 antibodies targeting cytochrome P450-2D6 and anti-liver cytosol (anti-LC) type 1 antibodies. Management frequently consists of lifelong nonspecific immunosuppression with azathioprine or other salvage therapies [3,4].

PBC usually affects women aged 40 to 60, with a male to female ratio of 1:10. It is characterized by the attack of the immune system towards the bile ducts and their epithelial cells, leading to progressive destruction of the intrahepatic bile ducts, and if not diagnosed and adequately treated it develops toward fibrosis, cirrhosis and finally liver failure [5]. Management consists of lifelong administration of ursodeoxycholic acid (UDCA) [6].

PBC and AIH can be difficult to distinguish clinically at early stages. There are still critical issues concerning early diagnosis, risk stratification of disease progression and identification of response to therapy predictors. Moreover, some patients may have “overlap syndromes” with characteristics of PBC or primary sclerosing cholangitis (PSC) in combination with AIH and cannot be assimilated into classical diagnostic categories. The most common AIH–PBC overlap syndrome [7] can be diagnosed using the Paris criteria, while so far there are no standardized diagnostic criteria for the other types of overlap syndromes [8]. In recent years, the definition of “overlap syndrome” has been substituted by the new definition of “variant syndrome of AIH and PBC” [9]. The variant syndrome shares diagnostic clinical and histological features of both entities [10]. Another challenge in AIH/PBC diagnosis is represented by the coexistence of comorbidities which may affect clinical phenotype at presentation. The most common association was found with other concurrent extrahepatic autoimmune disorders (CEHAID), mainly with autoimmune thyroid disease, but also with Sjögren’s syndrome, rheumatoid arthritis, nondestructive polyarthropathy, type 1 diabetes, vitiligo, ulcerative colitis and psoriasis [11,12].

AIH and PBC can be differentiated using clinical, biochemical, serologic and histologic findings. Blood testing is often the first step for the diagnosis of AILDs because many patients do not show symptoms until the disease has progressed to cirrhosis or liver failure. Serum autoantibodies ANA/SMA and LKM-1 are generally considered the diagnostic hallmarks of AIH [2]; however, a recent review reported a very poor diagnostic accuracy for ANA, SMA and LKM-1 if detected alone, with accuracy increasing only in the presence of both ANA and SMA [13]. Moreover, ANA tests also detect antigen specificities associated exclusively with PBC, including autoantibodies to Sp100-containing nuclear bodies (NBs) or gp210 protein [14]. Among candidate autoantibodies that may aid in the diagnosis of AIH, the most promising is α-actinin, a ubiquitous cytoskeletal cross-linking protein within the family of filamentous actin (F-actin) [15]. Nevertheless, a minority of patients with AIH do not show detectable autoantibodies at presentation and may express them only later and sporadically [2]. As a result, such patients must be scored using revised diagnostic criteria (RDC) or should undergo liver biopsy. Current guidelines recommend liver biopsy as a prerequisite for the diagnosis of AIH, to determine disease severity and to discriminate acute and chronic forms [16]. Regarding the diagnosis of PBC, liver biopsy is required only in specific anti-mitochondrial antibodies (AMA)-negative patients [17].

In this scenario, while the possible use of saliva as a diagnostic fluid has been largely investigated for oral and systemic diseases [18,19,20,21,22], it has been only marginally used in autoimmune liver diseases, mainly to investigate the role played by oral microbiota in their pathogenesis [23]. As a mirror of oral and systemic health, saliva provides valuable information because it contains not only proteins specifically secreted by the salivary glands [24], but also proteins from the gingival crevicular fluid [25,26], from oral microflora [27] and plasmatic proteins transported from blood to saliva by both intra- and extracellular pathways.

In a recent work, performed on the acid-soluble fraction of saliva from AIH and PBC patients explored by a top-down mass-spectrometry pipeline, our group evidenced qualitative and quantitative variations of some naturally occurring proteins/peptides belonging to the main families of salivary proteins, either secreted or not secreted by salivary glands [28]. These proteins/peptides were acid proline-rich proteins; statherin and P-B peptide; histatins; salivary cystatins; cystatins A, B, C and D; α-defensins; antileukoproteinase; S100A7, S100A8, S100A9 and S100A12 proteins. Supervised machine learning analysis of Mass Spectrometry (MS) data revealed for the first time the feasibility of selected salivary proteins to discriminate groups of subjects based on AIH or PBC occurrence. The acid-soluble salivary fraction did not contain, however, protein prone to precipitation under acidic conditions which may be influenced, in their expression, by the pathological condition. With the aim of extending our previous finding on salivary proteome in patients affected by AIH and PBC, and therefore to obtain a global overview of quantitative variations on the salivary proteome in presence of these two pathologies, we explored the acid-insoluble salivary fraction using a gel-based bottom-up proteomic approach combined with a robust machine learning statistical analysis of the dataset. Mass spectrometry data were analyzed by exact Mann—Whitney and Kruskal—Wallis tests to provide plausible salivary biomarkers related to AIH or PBC while Random Forest (RF) and multidimensional scaling (MDS) were used to individuate a panel of salivary proteins able to accurately classify the subjects based on AIH or PBC manifestation. Finally, a topology-based functional enrichment analysis has been performed for gaining insights into biological and functional pathways of both proteins commonly found and proteins showing varied levels among the three comparison groups.

## 2. Results

### 2.1. Characteristics of the Participants

A total of 17 AIH patients (in the following indicated as AIHp) 60.6 ± 12.8 (mean ± standard deviation) years old, males *n* = 2, females *n* = 15, and 17 PBC patients (PBCp) 63.0 ± 9.4 years old, males *n* = 0, females *n* = 17 were recruited. The control group (HCs) included 17 age- and sex-matched healthy volunteers (58.9 ± 13.4 years old, males *n* = 0, females *n* = 17).

The detailed demographic data of HCs, AIHp and PBCp are reported in Appendix A. Demographic characteristics, including age and gender, were matched between the AIHp and HCs and between PBCp and HCs (*p* > 0.05). Clinical and pharmacological features of AIHp and PBCp, collected at the same time of saliva sampling, are reported in Table 1. They comprise a panel of serum markers of liver dysfunction, including markers of hepatocellular dysfunction alanine transaminase (ALT) and/or aspartate aminotransferase (AST); markers of biliary disease such as alkaline phosphatase (ALP); markers of parenchymal liver dysfunction or biliary obstruction; or total bilirubin (TB) and γ-glutamyltransferase (GGT) and albumin, useful in assessing hepatic synthetic function. Patients were also tested for antinuclear antibodies (ANAs), smooth muscle antibodies (SMAs) and renal microsomal antigen antibodies (LKMs).

Regarding pharmacological therapies, 58.8% of AIHp were under Azathioprine only or in combination with Steroids and/or UDCA, 11.8% under Steroid only, 23.5% under UDCA only or with Steroids and 5.9% were without therapy (naïve); PBCp were 100% under UDCA treatment.

The presence of other concurrent autoimmune diseases was investigated in patients; five AIHp and six PBCp presented Hashimoto’s thyroiditis, one AIHp and one PBCp presented rheumatoid arthritis.

### 2.2. Saliva Sampling

To unravel the best approach for protein solubilization, the acid-insoluble fraction of saliva samples was subjected to three different solubilization solutions before bottom-up analysis. Based on bicinchoninic acid (BCA) assay, it was possible to demonstrate that 2% sodium dodecyl sulfate (SDS), 0.5 mM Dithiothreitol (DTT), 30 mM Tris-HCl pH 6.8 (Solution 3) allowed the retrieval of the best quantity of acid-insoluble proteins with a total protein concentration (TPC) of 2.5 mg/mL, approximately double the recovery of proteins compared to Solution 2 (TPC = 1.3 mg/mL) and more than 10 times the recovery obtained with Solution 1 (TPC = 0.2 mg/mL). Therefore, solution 3 was chosen to treat all AIHp, PBCp and HC samples.

Appendix A reports the stained SDS-PAGE gel of AIHp, PBCp and HC pellets. For each lane, three portions of the gel were manually excised and submitted separately to trypsin digestion. This approach was preferred over total lane digestion to minimize the amount of gel to be submitted to in-gel digestion and therefore improve protein identification.

### 2.3. RP-nanoHPLC-High Resolution ESI-MS and MS/MS Analysis and Protein Identification

MS analysis of tryptic peptides resulted in 918, 891 and 787 high-confidence proteins for AIHp, PBCp and HCs, respectively, of which 467 were quantified in all groups and used for statistical analysis. Mann—Whitney and Kruskal—Wallis tests were used to identify possible protein abundance variations among AIHp vs. PBCp, AIHp vs. HCs and PBCp vs. HCs. In this way, non-parametric tests highlighted 14 varied proteins out of 467 with significant changes (*p*-values < 0.05 with and FDR < 10%). The results are shown in Table 2, reporting: (i) six proteins decreased in AIHp compared to HCs, namely Arginase (P05089), Junction plakoglobin (P14923), Desmoplakin (P15924), Hexokinase-3 (P52790), Desmocollin-1 (Q08554) and Hornerin (Q86YZ3); (ii) one protein increased in AIHp compared to HCs: BPI fold-containing family A member 2 (Q96DR5); (iii) seven proteins increased in PBCp compared to HCs, namely Gelsolin (P06396), Monocyte differentiation antigen (P08571), Tumor-associated calcium signal transducer 2 (P09758), Clusterin (P10909), Heterogeneous nuclear ribonucleoproteins A2/B1 (P22626), Cofilin-1 (P23528) and BPI fold-containing family B member 2 (Q8N4F0); (iv) one protein decreased in PBCp compared to HCs: Hornerin (Q86YZ3). Interestingly, only Hornerin was found to be simultaneously reduced in both pathologies. No significant changes were found in the comparison between AIHp and PBCp.

### 2.4. Subjecs Classification with Random Forest (RF) Analysis

RF classification among AIHp, PBCp and HCs was applied to a subset of 17 proteins selected according to the Boruta method. Figure 1A shows the relative importance of the selected proteins for classification, evaluated by the mean decreased Gini purity score (*x*-axis) and the mean decreased accuracy (*y*-axis). These parameters account for how much the purity and accuracy of classification decrease by excluding the specific protein from classification. Therefore, the greater the value of the parameter the greater the importance of the protein in the model. In Figure 1, Hornerin (Q86YZ3) appears to be the most important protein, consistent with the fact that it is the only protein showing significant changes in both comparisons: AIHp vs. HCs and PBCp vs. HCs (Table 2). Clusterin (P10909) was also a good discriminant component.

The most important proteins for the classification of each group are shown in Figure 1B. As evidenced from this figure, Arginase (P05089), Desmoplakin (P15924), Cornifin-B (P22528), Heterogeneous nuclear ribonucleoproteins A2/B1 (P22626), 40S ribosomal protein S3 (P23396), 40S ribosomal protein S3a (P61247), 14-3-3 protein zeta/delta (P63104), Tubulin alpha-4A chain (P68366), Desmocollin-1 (Q08554) and Calnexin (P27824) were the most important proteins for AIHp classification, while Phosphoglycerate kinase 1 (P00558), Monocyte differentiation antigen (P08571), Clusterin (P10909) and Ras GTPase-activating-like protein (P46940) were selected for PBCp classification.

This subset of proteins provided a consistent increase in classification accuracy. The out-of-bag (OOB) classification error for AIHp and PBCp was 6% and 12%, respectively. When grouping AIHp and PBCp, the classification error of patients versus HCs was 0% (Table 3).

Figure 2 shows the multidimensional scaling (MDS) diagram obtained by RF classification analysis of the three compared groups. The MDS diagram highlights a complete separation between HCs and both AIHp and PBCp in accordance with 100% accuracy in the classification shown in Table 3. A 3D video showing the first three MDS axes is reported in Appendix A.

Seven proteins, selected by the Boruta algorithm with high MDG scores, showed also significant differences by Mann–Whitney tests. However, other components, prevalently those with low mean decreased Gini scores, did not show significant changes and, on the other hand, components with significant changes were not selected by the Boruta algorithm. This apparent contrast is due to the essential nature of RF classification and in general of methods based on decision trees. Indeed, RF is able to find multiple ‘split’ points of the same variable, a method completely different from that adopted by classical univariate tests that consider the whole distribution of data. This allows two or more groups to be discriminated by a certain variable even when the mean (or average rank) of the variable is the same in the different groups. On the other hand, the use of multiple split points is not very suitable for normal diagnostic purposes, which require values gradually and coherently related to the severity of the disease. Because of this fact, although the classification produced by RF is of considerable interest in several respects, the proteins that collectively contribute to the classification of AIHp and PBCp cannot be tout court considered as candidate markers of the diseases.

The relative abundance of 23 proteins, including the 14 proteins with significant changes based on Mann–Whitney tests and the 17 proteins selected by Boruta algorithm for RF classification (eight proteins were in common) are graphically shown in the heatmap of Figure 3. From the heatmap it is interesting to observe the higher values of Hornerin (Q86YZ3; last row) in the HCs group with respect to both AIHp and PBCp groups, in agreement with the results of the Mann–Whitney tests.

To assess the diagnostic value of Hornerin as a potential salivary biomarker of autoimmune liver diseases, Hornerin data were evaluated by a ROC curve, by setting HCs versus AIHp and PBCp grouped together (Figure 4). The area under the ROC curve (AUC) was 0.939, and the best cut-off resulted in a sensitivity and specificity of 82% and 74%, respectively. 

### 2.5. Protein-Protein Interaction Network, Topological Analysis and Pathway Enrichment

The topological features of the protein-protein interaction network (PPIn), such as node degree and betweenness centrality (BC) distributions, were examined using the Network Analyzer tool of Cytoscape. In scale-free networks, often observed in biological systems, most of the nodes (proteins) have few connections to the others, while a small number of nodes (hubs) are extensively connected to numerous others within the network. The nodes with high degrees were identified as hub proteins, which represent important connections within the network structure. On the other hand, nodes with high BC values were identified as bottleneck proteins because of their central role in facilitating communication and information flow between different parts of the network.

#### 2.5.1. Proteins Commonly Found among AIHp, PBCp and HCs

The extended PPIn obtained from the 467 proteins found among AIHp, PBCp and HCs, generated by the STRING database, is shown in Figure 5A.

The PPIn was composed by only one giant network containing 449 nodes, interacting with 2293 edges and having a PPI enrichment *p*-value < 1.0 × 10^−16^. Among the 449 nodes, 47 nodes were selected for their higher BC value and 47 nodes for their larger degree. The 25 nodes having the highest BC (min. 0.008; max. 0.132) and degree (min. 58; max 177) values, with respect to the full PPIn, were extracted from the extended network, and constitute the backbone network (Figure 5B). In the backbone network Glyceraldehyde-3-Phosphate Dehydrogenase (*GAPDH*) and albumin (*ALB*) were hub proteins with the largest degree and highest BC values. The other proteins of the backbone network were Cofilin (*CFL1*), T-Complex 1 protein (*TCP1*), Eukaryotic Translation Elongation Factor 2 (*EEF2*), Cell Division Cycle 42 (*CDC42*), Ras Homolog Family Member A (*RHOA*), Matrix Metallopeptidase 9 (*MMP9*), Integrin Subunit Alpha M (*ITGAM*), ATP Synthase F1 Subunit Alpha (*ATP5F1A*), ATP Synthase, F1 Subunit Beta (*ATP5B*), Valosin-Containing Protein (*VCP*), Pyruvate Kinase M1/2 (*PKM*), Heat Shock Protein Family A (*Hsp70*) Member 8 (*HSPA8*), Heat Shock Protein Family A (*Hsp70*) Member 5 (*HSPA5*), Heat Shock Protein Family D (*Hsp60*) Member 1 (*HSPD1*), Heat Shock Protein Family A (*Hsp90*) Member 1 (*HSP90AA1*), Heat Shock Protein Family B (*Hsp90*) Member 1 (*HSP90AA1*), Catalase (*CAT*), Calreticulin (*CALR*), Prohibitin 1 (*PHB1*), Enolase 1 (*ENO1*), Prolyl 4-hydroxylase Subunit Beta (*P4HB*) and Fibronectin 1 (*FN1*).

Finally, the 25 proteins included in the backbone network underwent GO biological process annotation and Reactome pathway analysis (Table 4).

#### 2.5.2. Proteins with Varied Levels among AIHp, PBCp and HCs

The PPIn built by using the 23 proteins obtained by merging those showing significant varied levels among AIHp, PBCp and HCs (14 proteins) based on Mann–Whitney tests, with the proteins selected by the Boruta algorithm (17 proteins), is shown in Figure 6.

The PPIn was composed of one main network containing 13 nodes, interacting with each other through 15 edges, and three minor networks (topological features of the nodes are reported in Appendix A). The protein with the highest BC and largest degree is Phosphoglycerate Kinase 1 (*PGK1*) followed by 14-3-3 protein zeta/delta (*YWHAZ*) and Cofilin-1 (*CFL1*). One minor network comprised three components of desmosomes, namely Junction plakoglobin (*JUP*), Desmoplakin (*DSP*) and Desmocollin-1 (*DSC1*); the second comprised two antimicrobial peptides, namely BPI fold-containing family A member 2 (*BPIFA2*) and BPI fold-containing family B member 2 (*BPIFB2*) and the last comprised two cytoskeleton components, namely Hornerin (*HRNR*) and Small Proline Rich Protein 1B (*SPRR1B*).

Finally, also these proteins underwent GO biological process annotation and Reactome pathway analysis as reported in Table 5.

## 3. Discussion

The aim of this study was to provide a salivary molecular signature of autoimmune liver diseases. A robust statistical analysis of quantitative salivary proteomic data from AIHp, PBCp and controls provided indication of the proteins statistically varied in levels among groups, considered as potential biomarkers of the disease, and proteins able to classify a subject with good accuracy based on AIH or PBC occurrence. Additionally, proteomic data have been used to build a PPIn and a functional enrichment analysis has been performed to explore the biological significance of salivary proteins and obtain insights regarding the contribution of these proteins to the pathogenesis of autoimmune liver diseases.

### 3.1. Topology-Based Functional Enrichment Analysis

In the backbone network, Glyceraldehyde-3-Phosphate Dehydrogenase (*GAPDH*) and albumin (ALB) were either hub and bottleneck proteins with the largest degree and BC values. It is reported that proteins that are hubs as well as bottlenecks will likely be evolutionarily conserved [29] and are involved in multiple pathways [30].

In the contest of chronic liver disease, both quantitative and functional changes have been evidenced in either albumin or *GAPDH* [31]. It has been demonstrated that in patients with liver cirrhosis, albumin undergoes both several reversible and irreversible posttranscriptional changes that alter its properties [32]. Importantly, albumin represented a good prognostic factor, being a significant predictor of death in patients with liver cirrhosis [33,34]. Ideally, topological network analysis identifies proteins susceptible to be potential biomarker of pathologies [35]; however, our proteomic analysis of saliva from AIH and PBC patients did not evidence varied levels of these two important proteins. Indeed, among the 25 proteins constituting the backbone network, and thus showing higher degree and BC values, only Cofilin-1 (degree 68, BC 0.016) was found to be increased in PBC patients with respect to HCs. As reported below, Cofilin-1 is a multifunctional protein involved in many biological processes and functional pathways also in the liver.

The topology-based functional enrichment analysis performed on the backbone’s proteins revealed a set of central nodes mainly associated with the immune system; among these Interleukin-4, Interleukin-13 and Interleukin-12 signaling displayed higher *p*-values. Hepatocytes, cholangiocytes, putative hepatobiliary progenitor cells and fibroblasts express functional interleukin-4 and interleukin-13 receptors, and studies conducted on transgenic mice with interleukin-13 signaling genetically disrupted in hepatocytes, cholangiocytes or fibroblasts revealed key roles for interleukin-13 in fibrosis, steatosis, cholestasis and ductular reaction [36]. PBC has a genetic association with interleukin-12 signaling [37], so that modulation of this pathway at an early stage of disease has been proposed as a therapeutic model in the treatment of this pathology [38]. Regarding proteins with varied levels among AIHp, PBCp and HCs, the highest central nodes were Phosphoglycerate Kinase 1 (*PGK1*) followed by 14-3-3 protein zeta/delta (*YWHAZ*) and Cofilin-1 (*CFL1*). The topology-based functional enrichment analysis performed on these salivary proteins highlighted an enrichment of terms with multifaceted biological functions mostly related to the immune system, but also with a strong involvement in the liver fibrosis process and with antimicrobial activity.

### 3.2. Dysregulated Proteins in AIH and PBC Are Mainly Involved in Liver Fibrosis

Liver fibrosis is an abnormal wound repair response caused by an assortment of chronic liver damages, which is characterized by over-deposition of diffuse extracellular matrix and anomalous hyperplasia of connective tissue [39]. Our analysis performed on the salivary proteome of AIH and PBC evidenced a set of dysregulated proteins with respect to HCs potentially involved in the development and/or progression of this pathological condition. These findings were further confirmed by the topology-based biological and pathways enrichment analysis that evidenced an enrichment of terms related not only to the immune system, but also to epithelium development, regulation of cell adhesion and formation of the cornified envelope.

The concomitant reduced level of Junction plakoglobin, Desmoplakin and Desmocollin-1 in AIH patients is intriguing since all three proteins are major components of desmosomes, cell structures specialized in cell-to-cell adhesion. The reduction in desmosomal protein level in AIHp, evidenced for the first time in this study, could be related to their liver fibrotic state; in fact, 70% of the AIHp enrolled in the study presented the most severe liver fibrotic stage III and IV while the majority of PBCp presented a mild liver fibrotic stage I and II. The exact localization of cell junctions in hepatic epithelial cells has not been determined yet [40], but Zhou et al. have shown that, in the absence of plakoglobin, bile duct ligation resulted in a more severe disease outcome with enhanced liver fibrosis [41]. The correlation between the levels of desmosomal proteins and the fibrotic stage has been described in mice lacking E-cadherin in the liver which developed periportal inflammation via an impaired intrahepatic biliary network, as well as periductal fibrosis, which resembles primary sclerosing cholangitis [42]. These findings point out that an intact intrahepatic biliary network with normal bile secretion depends on functional E-cadherin. Moreover, Dubash et al. described how the loss of desmosomal proteins in transgenic mice activated a signaling pathway in cardiomyocytes that up-regulated multiple inflammatory and extracellular matrix proteins known to promote tissue fibrosis [43].

Arginase-1 levels were found decreased in AIHp with respect to HCs and important for AIH classification in RF analysis. This protein is a manganese-containing enzyme that catalyzes the final step in the urea cycle. In the liver arginase-1 is mainly expressed in the periportal hepatocytes, but not in the bile ducts, endothelial and Kupffer cells [44]. Serum arginase level is significantly associated with oxidative stress since it is indirectly involved in nitric oxide (NO) regulation by the consumption of L-arginine, which is a common substrate for NO synthase (NOS). The balance between the consumption of L-arginine by arginase-1 (leading to the production of proline for collagen and polyamine production which are essential for cell growth and matrix modelling) and NOS (for NO production) determines the outcome of wound repair, with arginase-1 controlling the healing process and NOS regulating the anti-microbial activity. In a mouse model of liver fibrosis, hepatic stellate cells activation is accompanied by a switch in arginine catabolism resulting from downregulation of NOS and upregulation of arginase-1; therefore, inhibition of arginase-1 has been proposed as an anti-fibrotic target for the treatment of liver fibrosis [45]. Our results, which evidenced reduced arginase-1 levels in AIHp, are apparently in contrast with these findings, however it can be highlighted that most of the patients enrolled for this study were undergoing corticosteroids therapy whose effect on arginase-1 expression remains to be elucidated [46].

Gelsolin and Cofilin-1, both showing increased levels in PBCp with respect to HCs, are actin-binding proteins, key regulators of actin filament assembly and disassembly. In particular, Cofilin-1 is, among the proteins found with varied levels in AIHp and PBCp, the only one included in the backbone network probably thanks to its prominent role in actin dynamics and modulation essential for cell survival. This prominent role in the topological analysis is not surprising since liver injuries create a microenvironment that alters actin dynamics in the hepatic stellate cells which are responsible for type I collagen expression, the major extracellular matrix protein in various types of fibrotic diseases [47]. Hereafter, Cofilin-1 levels were related to the stage of liver fibrosis playing a key role in the progression of liver fibrosis toward hepatocellular carcinoma [48]. Moreover, Cofilin-1 levels have been found to be increased in HBV-associated hepatocellular carcinoma and its levels were correlated with the severity of this liver disease [49].

Early experiments on mice lacking gelsolin evidenced a reduction in dermal fibroblasts migration, establishing that gelsolin is required for rapid motile responses in cell types involved in stress responses such as hemostasis, inflammation and wound healing [50].

Similarly, clusterin, a Golgi extracellular chaperone implicated in cholestatic and fibrotic processes, was also found to be augmented in PBC patients with respect to HCs. It is expressed by hepatocytes and secreted into the bile, possibly acting as a chaperone to protect either bile duct epithelia from damage by toxic bile constituents or biliary proteins from misfolding [51]. Clusterin was found to be upregulated in thioacetamide-induced and bile duct ligation mouse models of liver fibrosis; the upregulation of clusterin attenuated hepatic fibrosis by inhibiting the hepatic stellate cells’ activation and Smad3 signaling pathways responsible for the production of extracellular matrix proteins such as type I collagen [52]. On the contrary, circulating clusterin levels were significantly reduced in biliary atresia, which is a rare cholestatic liver disease of neonates characterized by obstruction of the biliary system; in this case clusterin levels were reduced, especially in patients with worse outcomes including jaundice and severe liver fibrosis [53]. Our results, in agreement to these findings, suggested that upregulation of clusterin may act as a defense mechanism to prevent hepatic fibrosis.

Higher levels of Tumor-associated calcium signal transducer 2 (Trop-2) were found in PBC patients with respect to HCs. It has been reported that Trop-2 levels in hepatitis C patients were inversely correlated to AST and ALT values [54], but our results cannot confirm this correlation since PBC patients enrolled for the study presented almost normal values of transaminases. Trop-2 plays a role in tumor progression by actively interacting with several key molecular signaling pathways traditionally associated with cancer development and progression. Conversely to most type of tumor, where Trop-2 was found upregulated [55], in liver cancer this protein has been found downregulated [56]. The different expression pattern of Trop-2 in liver cancer and autoimmune liver diseases is intriguing and deserves further elucidation.

The same controversial results in the expression pattern between liver cancer and autoimmune liver diseases has been highlighted for hornerin; this protein was found to be elevated via proteomic analysis in hepatocellular carcinoma [57] while our proteomic data revealed hornerin as the only protein with levels reduced in both AIH and PBC patients with respect to HCs as well as the protein that, in RF analysis, best discriminated these pathologies from HCs. This contradiction is quite interesting and could be related to different pathophysiological pathways between cancer and AIH and PBC. For instance, hornerin was found to be downregulated in atopic dermatitis, contributing to the epidermal barrier defect observed in this skin disease [58]. Indeed, being a component of the liver matrisome [59], it can be speculated that reduced levels of hornerin in both AIH and PBC patients may be correlated to the anomalous hyperplasia of connective tissue associated with liver fibrosis. The role of hornerin as a potential biomarker for AIH/PBC classification has been further validated by the ROC curve of HCs versus AIHp and PBCp grouped together, which provided an AUC of 0.93 with an accuracy of 78%. However, it may be highlighted that a better classification accuracy of HCs of 100% with respect to AIHp and PBCp grouped together was obtained by RF using the whole panel of 17 proteins selected by the Boruta algorithm.

Our study on the salivary proteome of AIH and PBC revealed also varied levels of proteins with prevalent antimicrobial function, and marginally involved in the liver fibrosis process. Indeed, several studies have shown that metabolites of oral microbiota, through the gut-oral axis, can enter the bloodstream and contribute to the occurrence and progression of many liver diseases [60,61].

BPI fold-containing family A member 2 (*BPIFA2*) and BPI fold-containing family B member 2 (*BPIFB2*) proteins are characterized by the presence of the bactericidal/permeability-increasing protein fold (BPI fold) [62].

Interestingly, despite their similar antimicrobial function, increased levels of BPIFA2 and BPIFB2 were observed in AIH and PBC patients, respectively, with respect to HCs. This difference could be related to an oral dysbiosis, which is often observed in such patients and involves the alteration of different bacterial species in the oral microbiota of AIHp [23] and PBCp [63] and towards which *BPIFA2* and *BPIFB2* can show different inhibitory activity. Indeed, a recent study on *BPIFA2* knockout mice highlighted its prominent role in the solubilization of ingested bacterial lipopolysaccharides [64]. From another point of view, the increased level of *BPIFA2* in AIHp only could be related to other biological functions of this multifaceted protein. In fact, besides its prominent role in the local antibacterial response, *BPIFA2* exerted a role in the prevention of early fibrosis progression and the development of chronic kidney disease [65]. A similar protective role in inhibiting liver fibrosis can be speculated for the AIHp since the patients enrolled for this study showed a marked liver fibrotic state.

An impaired gut-oral axis in autoimmune hepatitis could be also related to the increased level of Monocyte differentiation antigen CD14 we observed in PBCp with respect to HCs. CD14 might be an important factor in the pathogenesis of PBC since it is constitutively expressed by human intrahepatic biliary epithelial cells [66]. As a marker of monocyte activation and response to bacterial lipopolysaccharides, CD14 level reflects the host response to products of microbial translocation [67]. Increased levels of CD14 were found in PBC patients before UDCA therapy [68], but our results showed an increase in CD14 levels also in patients undergoing therapy with UDCA. CD14 protein also plays a key role in liver disease by reducing fibrosis through degradation or clearing up collagen-I deposits [69].

### 3.3. Machine Learning Analysis of Proteomics Data

RF analysis, one of several up-to-date machine learning methods, evidenced a panel of proteins/peptides present in saliva able to correctly classify the AIHp group with respect to PBCp with 94% accuracy, the PBCp group with respect to the AIHp group with 88% accuracy and PBCp/AIHp grouped together with respect to HCs with 100% accuracy. This result confirms our previous one obtained by applying the same statistical approach to a mass-spectrometry dataset obtained after top-down proteomic analysis of the acid-soluble fraction of saliva from AIHp and PBCp. Moreover, this approach confirms the feasibility of the salivary proteome to discriminate groups of subjects based on physiological or pathological condition not only confined to the oral cavity [28].

### 3.4. Study Limitation

The low number of subjects involved in the study may not be entirely representative of the considered population; nevertheless, the statistical analysis performed by our group provides a good classification of subjects based on AIH or PBC occurrence. A larger population will be useful to further validate the present findings. Most of the proteins found with altered levels among AIHp, PBCp and HCs and/or selected by Boruta algorithm for patients’ classification have multiple isoforms often characterized by different, sometimes even opposite, functional activities, and the bottom-up approach exploited in this study does not allow for characterization of isoforms/proteoforms.

Moreover, the observational nature of the study did not allow a demonstration of the causal effect of the varied proteins on AILDs, which can be considered potential biomarkers rather than causal mediators of these pathologies.

## 4. Materials and Methods

### 4.1. Ethical Statement

This is a cross-sectional study performed in 2021 on AIHp and PBCp recruited from the liver unit of University Hospital of Cagliari, Sardinia, Italy. Patients and healthy controls signed the informed written consent that agreed with the latest stipulations established by the Declaration of Helsinki. The Committee of the “Azienda Ospedaliero-Universitaria di Cagliari”, Cagliari, Italy, approved the study on 21 July 2021 (reference number PG/2021/11303).

### 4.2. Study Subjects and Clinical Studies

Patients were diagnosed based on the criteria reviewed by the International Autoimmune Hepatitis Study Group (IAIHG) in 1999 [70] and by the EASL clinical practice guidelines [5]. The study included patients showing, at the time of saliva sampling, almost normal values of ALT and AST; based on these inclusion criteria, only patients that were under pharmacological therapy for at least three years were selected. Only one AIHp was without therapy but was included in this study because of his normal values of transaminases. Patients affected by overlap syndrome, chronic hepatitis induced by HBV or HCV, drug or alcohol abuse, fatty liver disease, primary sclerosing cholangitis and any major oral disease (periodontitis, caries) were excluded.

The HCs included age- and sex-matched healthy volunteers recruited from the local population. Controls were excluded if they were relatives of the patients and had a history of liver diseases, immunological disorders and major oral diseases. Most of the controls were patients’ caregivers recruited in the hospital during follow-up and/or medical and research personnel involved in the study.

### 4.3. Sample Collection, Treatment and Acid-Insoluble Proteins Solubilization

Unstimulated whole saliva (WS) (from 0.2 to 1 mL) was collected with a soft plastic aspirator at the basis of the tongue from 9.00 am to 1.00 pm. in fasting conditions using a standard protocol optimized to preserve salivary proteins from proteolytic degradation. After collection, samples were immediately mixed with an equal volume of 0.2% (*v*/*v*) 2,2,2-trifluoroacetic acid (TFA) and centrifuged at 14,000× *g* for 10 min at 4 °C. The insoluble fraction (pellet) was separated and stored at −80° until the solubilization.

Three different solutions were tested for pellet solubilization: (i) Solution 1, 2% SDS, 0.4 M sodium chloride (NaCl); (ii) Solution 2, 0.1 M sodium hydroxide (NaOH); (iii) Solution 3, 2% SDS, 0.5 mM DTT, 30 mM Tris-HCl pH 6.8. To improve solubilization, the three solutions were submitted to three cycles of 1 min sonication/1 min vortex followed by 5 min centrifugation 14,000× *g*, 4 °C. The resulting pellet was submitted to a further solubilization step under the same conditions. The insoluble material was discarded, and the soluble fraction submitted to TPC determination in duplicate by Pierce^TM^ BCA Protein Assay kit (Thermo Fisher Scientific, Waltham, MA, USA), according to the provided kit instructions.

### 4.4. SDS-PAGE, Bands Excision and Enzymatic Digestion

All chemicals and reagents were purchased from Bio-Rad (Hercules, CA, USA). Ten μg of proteins were loaded on SDS-PAGE under reducing conditions. Samples were previous treated with Laemmli Buffer [71] for 5 min at 100 °C. The electrophoretic separation, carried out at 180 Volt constant for 30 min, was performed using 4–15% Mini-PROTEAN^®^ TGX™ Precast Protein Gels and Tris/Glycine Running Buffer (0.025 M Tris, 0.192 M Glycine, 0.1% SDS, pH 8.3). Molecular weights were determined by loading Precision Plus Dual Color Protein Standard. After the electrophoretic separation, gels were stained with Bio-Safe™ Coomassie Stain following the provided instructions. After destaining, each lane of the gel was divided into three slices corresponding to the following molecular weights: 250–75 kDa (A), 75–25 kDa (B) and < 25 kDa (C) (Appendix A), and slices, manually excised, cut into small pieces and submitted to in-gel digestion. Trypsin (Trypsin Singles, Proteomics Grade— Sigma-Aldrich/Merck, Darmstadt, Germany) was added to gel samples following the provided instructions in enzyme/proteins ratio of 1/80 (*w*/*w*) and incubated overnight at 37 °C. Extracted tryptic peptides were lyophilized and then solubilized in 0.1% formic acid (FA) for nano-RP-HPLC-high resolution ESI-MS and MS/MS analysis.

### 4.5. Nano-RP-HPLC-High Resolution ESI-MS/MS Analysis

All chemicals and reagents were purchased from Sigma-Aldrich/Merck. One hundred fifty-three samples corresponding to the tryptic peptides prepared by in-gel digestion of A, B and C slices obtained in triplicate from the acid-insoluble fraction of 17 saliva samples were analyzed, from September 2021 to November 2022, with an Ultimate 3000 Nano System HPLC (Dionex-Thermo Fisher Scientific) coupled with a LTQ Orbitrap Elite (Thermo Fisher Scientific). The Easy Spray reverse-phase nano column (250 mm × 75 μm inner diameter I.D., Thermo Fisher Scientific) was a C18 with 2 μm beads and elution of peptides was achieved with aqueous solvent A (0.1% FA) and aqueous solvent B (0.1% FA, 80% ACN *v*/*v*) in 100 min at a flow rate of 0.3 μL/min with the following gradient: 0–3 min at 4%B, 3–70 min 4–50% B, 70–90 min 50–80%B, 90–92 min 80–90%B, 92–100 min 90%B. The mass spectrometer was operating at 1.7 kV in the data-dependent acquisition mode, with the capillary temperature set at 275 °C and S-Lens RF level 68.4%. Full MS experiments were performed in positive ion mode from 350 to 1600 *m*/*z* with resolution 120,000 (at 400 *m*/*z*). The 10 most intense ions were subjected to CID fragmentation setting 35% of normalized collision energy for 10 ms, isolation width of 2 *m*/*z* and activation q of 0.25.

### 4.6. Protein Identification and Quantitation

Protein characterization was performed by Proteome Discoverer (PD) software (version 2.2, Thermo Fisher Scientific) with the SEQUEST HT cluster search engine (University of Washington, licensed to Thermo Electron Corporation, San Jose, CA, USA) against the UniProtKB Homo sapiens database (188,453 entries, release 2019_03). MS spectra obtained from the three gel portions of the same sample were analyzed by PD as fractions of that sample. Database search parameters were as follows: carbamidomethylation of cysteine as fixed modification, oxidation of methionine, serine/threonine phosphorylation, N-terminal pyroglutamic residue and N-terminal acetylation as dynamic modifications and allowance for up to two missed tryptic cleavages. The peptide mass tolerance was set to 10 ppm and fragment ion mass tolerance was 0.6 Da. Peptides were filtered for high confidence and a minimum length of 6 amino acids; settings of FDR were 0.01 (strict) and 0.05 (relaxed).

Proteins identified were then filtered for high FDR confidence and for a minimum number of 2 unique peptides, excluding keratins. Grouping and quantification were set specifying categorical factor related to the condition AIHp, PBCp and HC. Proteins identified have been subjected to PD Label-Free Quantification and protein abundances have been determined based on area of unique peptide precursor ions. The mass spectrometry proteomics data have been deposited with the ProteomeXchange Consortium via the PRIDE [72] partner repository with the dataset identifier PXD039847.

### 4.7. Data Analysis

Only proteins found with at least 30% of distribution among all the three comparison groups (AIHp, PBCp and HCs) were selected for statistical analysis. Protein abundances were first automatically normalized against the total amount of tryptic peptides by PD software, then transformed to log_2_ and submitted to quantile normalization. Values under the detection limit were replaced with log_2_(1000), a value below the minimum of the entire dataset. The choice of this conventional value was not critical for statistical analyses, as non-parametric tests and RF do not consider the continuous data distribution but only the rank order of values.

### 4.8. Protein-Protein Interaction Network, Topological Analysis and Pathway Enrichment

Protein–protein interaction (PPI) analysis was performed on 467 proteins found with at least 30% of distribution among all the three comparison groups (AIHp, PBCp and HCs) and on the 23 proteins obtained by merging those showing significant varied levels among AIHp, PBCp and HCs (14 proteins) based on Mann–Whitney tests, with the proteins selected by Boruta algorithm (17 proteins). Analysis was performed by STRING v.11.5 [73] (latest access on July 2023) with a default medium confidence of 0.4 and FDR stringency at 5%. Active interaction sources were based on “experiments”, “co-expression” and “co-occurrence”. Topological parameters of the PPI network, such as degree and betweenness centrality (BC) were calculated from Cytoscape v.3.10 [74]. The stringApp for Cytoscape was used to retrieve PPI networks from the STRING database [75] and topological analyses of the networks were performed using the Network Analyzer tool [76]. Two important metrics—degree and betweenness—were utilized to evaluate the importance of nodes in a network [77]. Hub proteins were identified by their very high degree of connectivity. Proteins with high betweenness centrality, namely bottlenecks, are key connectors in the PPI network, controlling the flow of information within a network [78].

For accessing the key nodes in the PPI, network members were first ranked by their degrees and BC afterwards the top-scoring proteins corresponding to 10% of the total number of nodes were selected. These topologically central proteins comprised the backbone of the PPIn. Selected proteins were further submitted to functional enrichment analysis based on reactome pathway distribution and GO biological process by stringApp, after removing redundant terms with default cutoff of 0.5.

### 4.9. Statistical Analysis

Various goodness-of-fit tests (Kolmogorov–Smirnov, Shapiro–Wilk, Lilliefors, etc.) showed a considerable deviation from normality of protein abundances. Thus, comparisons among the three groups were performed by Mann–Whitney (MW) and Kruskal–Wallis (KW) non-parametric tests. Significant *p*-values of multiple tests were adjusted by the Benjamini–Hochberg procedure [79] to keep the cumulative false discovery ratio (FDR) among all tests less than 10%. Classification of subjects was obtained using Random Forest (RF) analysis and optimized by selecting a subset of relevant proteins identified by the Boruta method [80] and by tuning the two main RF parameters (total number of trees and number of features randomly sampled for each split point). RF classification was validated by the ‘out-of-bag’ samples. In detail, this method consists in using only about two-thirds of the samples for each decision tree. The classification obtained with these samples is then tested using the remaining one-third of the samples (hence the term ‘out-of-bag error’). This procedure is repeated for each of the planned number of trees (1000 in our analysis), each time randomly selecting the samples for classification and those for validation. The overall accuracy is ultimately assessed as the average of the ‘out-of-bag’ errors. The relative importance of each protein for classification was evaluated by the decrease in the Gini purity index and decrease in classification accuracy observed after temporarily excluding that protein from the analysis. Dimensionally reduced diagrams of the RF classification were obtained by multidimensional scaling (MDS) using the RF proximity between each two subjects. Proximity was calculated as the normalized frequency of trees containing the two subjects in the same end node. Statistical analyses were made using R (R Core Team. R: A language and environment for statistical computing. Vienna, Austria: R Foundation for Statistical Computing; 2014. http://www.R-project.org/; latest access on 1 December 2022)

## 5. Conclusions

The bottom-up proteomic approach applied in this study, associated with a robust statistical analysis, allowed us to highlight a set of potential salivary biomarkers of AIH and PBC. The topology-based functional enrichment analysis performed on these potential salivary biomarkers highlighted an enrichment of proteins with multifaceted biological functions mostly related to the immune system, but also with strong involvement in the liver fibrosis process and with antimicrobial activity. Cofilin-1 was the protein with the highest centrality values in the backbone PPI network among those with varied levels, while Hornerin was the only protein showing reduced levels in both AIHp and PBCp compared to HCs and also able to better discriminate the patients of both groups from HCs in the RF analysis. RF analysis confirmed the feasibility of the salivary proteome to discriminate groups of subjects based on AIH or PBC occurrence as previously suggested by our group.

## Figures and Tables

**Figure 1 ijms-24-12207-f001:**
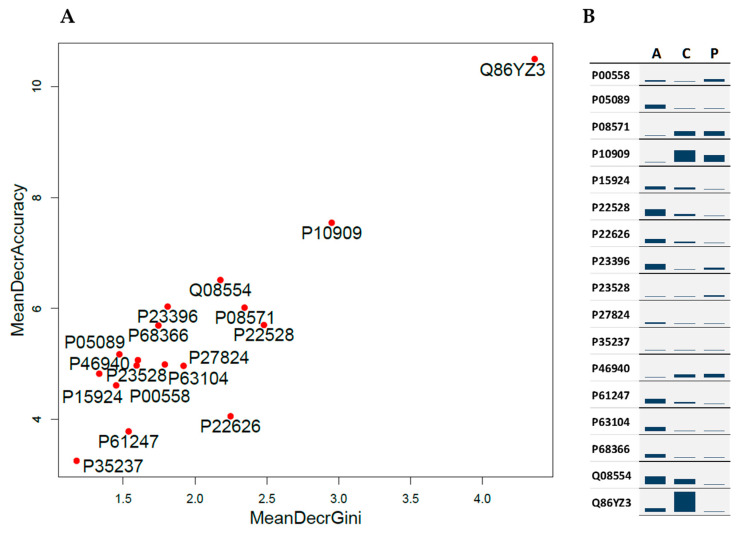
Variable importance for RF classification (**A**) Plot of the mean decreased Gini purity score (*x*-axis) and the mean decreased accuracy (*y*-axis). (**B**) Plot of the mean decreased Gini purity score (relative scale) calculated for each group. A, C and P are abbreviations of AIHp, HCs and PBCp, respectively.

**Figure 2 ijms-24-12207-f002:**
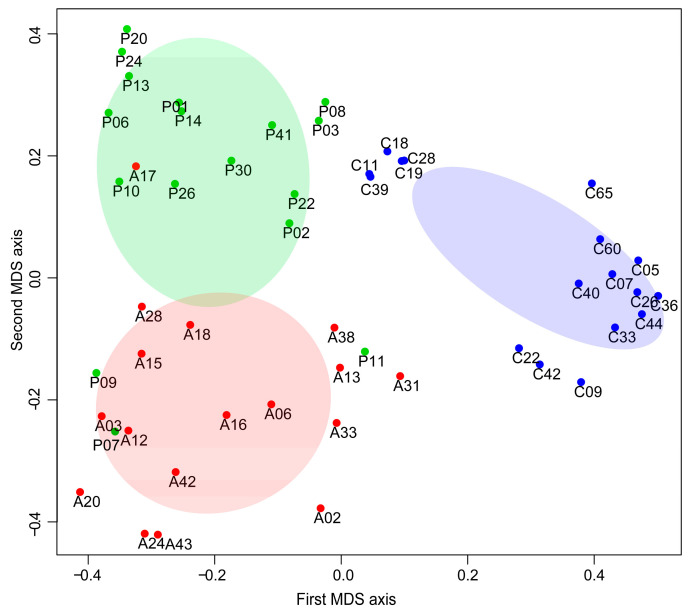
MDS diagram of RF classification, obtained by using the proximity between each pair of samples. A, C and P are abbreviations of AIHp, HCs and PBCp, respectively.

**Figure 3 ijms-24-12207-f003:**
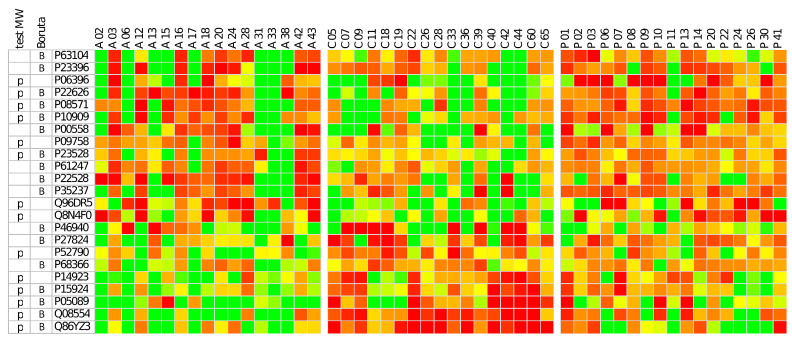
Heatmap of the abundance of proteins that showed significant changes based on M–W tests (denoted by the P code in the first column) and the proteins selected by Boruta algorithm for classification of AIH, PBC and HC groups (denoted by the B code in the second column). A, C and P are abbreviations of AIHp, HCs and PBCp, respectively. Eight proteins were in common in the two lists. To facilitate comparisons, data were row-wise standardized, that is each protein has mean = 0 and standard deviation = 1. Colors range from bright green (mean − 1 SD) to bright red (mean + 1 SD). The last row represents Hornerin protein, the only one that was significantly decreased both in AIHp and PBCp with respect to HCs.

**Figure 4 ijms-24-12207-f004:**
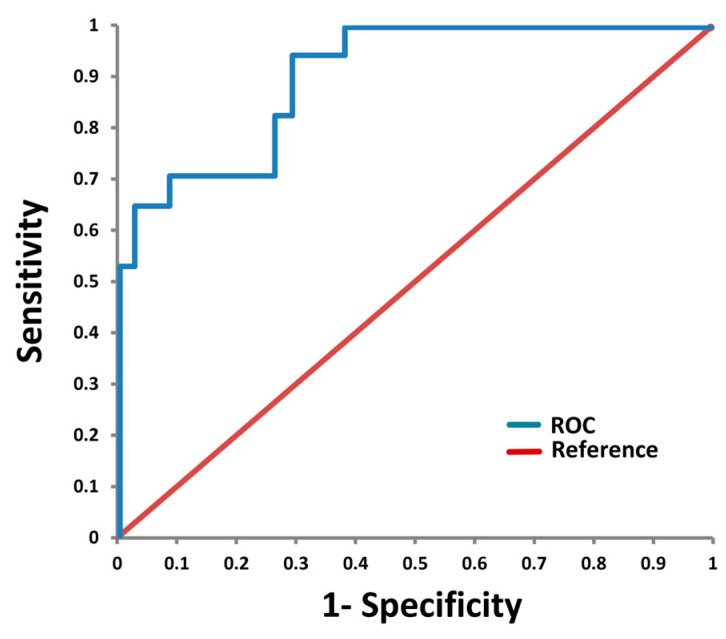
Receiver Operating Characteristic (ROC) curve for Hornerin (Q86YZ3) abundance in HCs versus both forms of AIHp and PBCp, grouped together with an AUC = 0.939.

**Figure 5 ijms-24-12207-f005:**
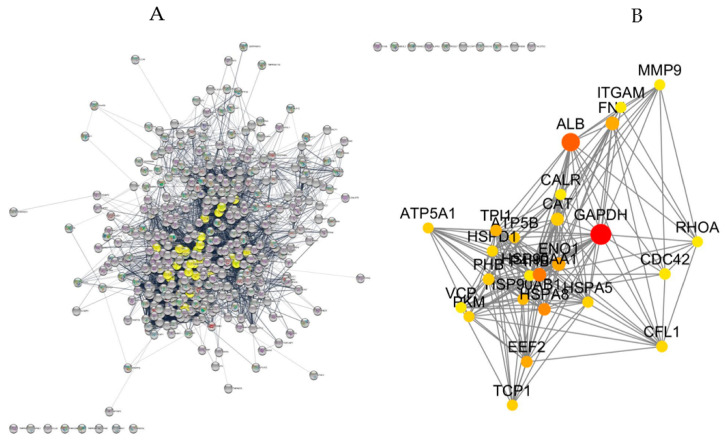
(**A**) Extended PPI network composed by 449 nodes interacting each other with 2293 edges and a PPI enrichment *p*-value < 1.0 × 10^−16^. Yellow nodes denote proteins with high degrees and betweenness centrality values thus representing the backbone network. (**B**) Topology of the backbone network, extracted from the extended network, containing proteins with higher BC and degree values. The size of the node corresponds to their BC values and colors range from yellow to red based on degree values. Protein names are reported in the text.

**Figure 6 ijms-24-12207-f006:**
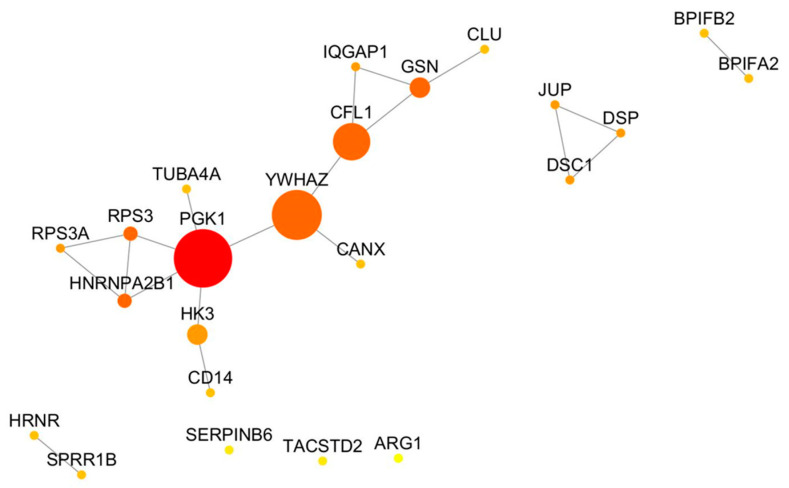
Topology of the network containing varied proteins among AIHp, PBCp and HCs. The size of the node corresponds to their BC values and colors range from yellow to red based on degree values. BC: Betweenness centrality.

**Table 1 ijms-24-12207-t001:** Clinical features and pharmacological treatments of AIHp and PBCp included in the study measured at the time of saliva sampling.

Parameters		AIHp	PBCp
Age, average (range)	Years	60.6 (40–83)	63.0 (52–83)
Gender, *n* (%)	Female	15 (88.3%)	17 (100%)
BMI, average (range)	Kg/m^2^	26.46 (17.58–36.13)	24.63 (19.33–30.27)
Cirrhosis, *n* (%)		4 (23.5%)	4 (23.5%)
Histological stage *n* (%)	I–II	5 (29.4%)	11 (64.7%)
	III–IV	12 (70.6%)	6 (35.3%)
Positivity to autoantibodies, *n* (%)	ANA	12 (70.6%)	8 (47.1%)
	SMA	10 (58.8%)	3 (17.6%)
	LKM	1 (5.9%)	2 (11.8%)
	AMA	0 (0%)	16 (94.1%)
AST, median (range)	IU/L	25.0 (14–57)	28.0 (17–65)
ALT, median (range)	IU/L	20.0 (8–46)	28.0 (17–133)
GGT, median (range)	IU/L	30.0 (0–138)	68.0 (14–452)
ALP, median (range)	IU/L	64.5 (28–216)	118.0 (65–415)
IgG, median (range)	g/dL	1.4 (0.69–2.51)	1.4 (1.1–2.3)
Albumine, median (range)	g/dL	3.9 (3.24–4.76)	3.9 (2.8–4.4)
Prothrombin time, median (range)	INR	0.93 (0.92–1.03)	1.0 (0.91–1.61)
TB, median (range)	mg/dL	0.69 (0.3–1.46)	0.61 (0.28–2.95)
Platelets, median (range)	10^9^/L	199.0 (91–402)	223 (46–418)
Pharmacological treatment (% treated)	AZA (only or + UDCA and/or Steroids)	58.8%	
	Steroids	11.8%	
	UDCA (only or + steroids)	23.5%	100%
	Naïve	5.9%	

Range refers to total range; BMI: body mass index; ANA: antinuclear antibodies, SMA: smooth muscle antibodies; LKM: renal microsomal antigen antibodies; AMA: anti-mitochondrial autoantibodies; AST: aspartate aminotransferase; ALT: alanine aminotransferase; GGT: glutamyl transferase; ALP: alkaline phosphatase, IgG: immunoglobulin G; TB: total bilirubin; AZA: Azathioprine; UDCA: Ursodeoxycholic Acid.

**Table 2 ijms-24-12207-t002:** Pairwise Mann–Whitney comparisons and Kruskal–Wallis test between HCs, AIHp and PBCp. Significant *p*-values < 0.05 with and FDR < 10% are highlighted. Direction of significant changes is also shown.

Proteins		HCs vs. AIHp	HCs vs. PBCp	AIHp vs. PBCp	HCs vs. AIHp vs. PBCp
Uniprot Code	Description	Mann Whitney	Mann Whitney	Mann Whitney	Kruskal Wallis
		*p*-Value	*p*-Value	*p*-Value	*p*-Value
P05089	Arginase *	0.0003 AIH < HC	0.0730	0.0061	0.0006
P06396	Gelsolin	0.2313	0.0015 PBC > HC	0.0653	0.0077
P08571	Monocyte differentiation antigen *	0.0256	0.0001 PBC > HC	0.1474	0.0005
P09758	Tumor-associated calcium signal transducer 2	0.0187	0.0010 PBC > HC	0.5352	0.0035
P10909	Clusterin *	0.0540	0.0002 PBC > HC	0.0777	0.0009
P14923	Junction plakoglobin	0.0007 AIH < HC	0.2313	0.0287	0.0034
P15924	Desmoplakin *	0.0007 AIH < HC	0.3057	0.0177	0.0030
P22626	Heterogeneous nuclear ribonucleoproteins A2/B1 *	0.0036	0.0006 PBC > HC	0.1899	0.0009
P23528	Cofilin-1 *	0.0679	0.0007 PBC > HC	0.5177	0.0062
P52790	Hexokinase-3	0.0013 AIH < HC	0.0987	0.0819	0.0053
Q08554	Desmocollin-1 *	<0.0001 AIH < HC	0.0064	0.0114	<0.0001
Q86YZ3	Hornerin *	<0.0001 AIH < HC	<0.0001 PBC < HC	0.4824	<0.0001
Q8N4F0	BPI fold-containing family B member 2	0.0145	0.0007 PBC > HC	0.5629	0.0035
Q96DR5	BPI fold-containing family A member 2	0.0007 AIH > HC	0.0216	0.5861	0.0044

* Proteins selected by Boruta algorithm for RF classification.

**Table 3 ijms-24-12207-t003:** Confusion matrix of the classification of the three groups of subjects.

		Predicted Class			
		HCs	AIHp	PBCp	OOB Error (%)
**True class**	HCs	17	0	0	0
	AIHp	0	16	1	6
	PBCp	0	2	15	12

**Table 4 ijms-24-12207-t004:** Top five most significantly enriched Gene Ontology (GO) Biological Process and Reactome Pathway enrichment terms for the proteins in the backbone network. Protein names are reported in the text.

Term ID	Term	Term Size	Enriched Terms	FDR *	Associated Genes
**Gene Ontology Biological Process**					
GO:0071310	Cellular response to organic substance	2369	21	5.99 × 10^−12^	*GAPDH|CAT|ATP5B|EEF2|TCP1|PKM|CALR|HSPA5|P4HB|HSP90AA1|FN1|VCP|HSP90AB1|MMP9|HSPD1|CDC42|RHOA|HSPA8|CFL1|ITGAM|PHB*
GO:0006457	Protein folding	213	9	6.53 × 10^−09^	*TCP1|CALR|HSPA5|P4HB|HSP90AA1|VCP|HSP90AB1|HSPD1|HSPA8*
GO:0046034	ATP metabolic process	204	8	1.12× 10^−07^	*GAPDH|TPI1|ENO1|ATP5B|PKM|VCP|ATP5A1|HSPA8*
GO:0043312	Neutrophil degranulation	484	10	1.42 × 10^−07^	*CAT|EEF2|PKM|HSP90AA1|VCP|HSP90AB1|MMP9|RHOA|HSPA8|ITGAM*
GO:0051702	Interaction with symbiont	93	6	7.16 × 10^−07^	*GAPDH|FN1|HSPD1|HSPA8|CFL1|PHB*
**Reactome pathways**					
HSA-168256	Immune System	1956	17	1.13 × 10^−08^	*CAT|EEF2|TCP1|PKM|CALR|HSPA5|P4HB|HSP90AA1|FN1|VCP|HSP90AB1|MMP9|CDC42|RHOA|HSPA8|CFL1|ITGAM*
HSA-3371556	Cellular response to heat stress	89	5	4.23 × 10^−05^	*HSPA5|HSP90AA1|VCP|HSP90AB1|HSPA8*
HSA-5336415	Uptake and function of diphtheria toxin	6	3	4.76 × 10^−05^	*EEF2|HSP90AA1|HSP90AB1*
HSA-6785807	Interleukin-4 and Interleukin-13 signaling	107	5	7.35 × 10^−05^	*HSP90AA1|FN1|MMP9|HSPA8|ITGAM*
HSA-9020591	Interleukin-12 signaling	46	4	1.10 × 10^−04^	*TCP1|P4HB|CDC42|CFL1*

* Corrected with Benjamini–Hochberg, Term size: the total number of proteins in the GO and Reactome pathways; Enriched terms: number of terms pertaining to the backbone network; Associated genes: proteins associated to the specific term.

**Table 5 ijms-24-12207-t005:** Gene Ontology (GO) Biological process and Reactome Pathway enrichment terms for the proteins varied among AIHp, PBCp and HCs.

Term ID	Term	Term Size	Enriched Terms	FDR	Associated Genes
**Gene Ontology Biological Process**					
GO:0045055	Regulated exocytosis	697	12	5.56 × 10^−08^	*TUBA4A|DSC1|IQGAP1|HK3|CD14|CLU|ARG1|HRNR|GSN|DSP|JUP|SERPINB6*
GO:0044419	Interspecies interaction between organisms	1899	12	1.60 × 10^−04^	*BPIFB2|CANX|BPIFA2|IQGAP1|RPS3|CD14|CLU|RPS3A|HNRNPA2B1|ARG1|GSN|CFL1*
GO:0060429	Epithelium development	1109	9	0.0011	*DSC1|IQGAP1|SPRR1B|HRNR|TACSTD2|PGK1|DSP|JUP|CFL1*
GO:0070268	Cornification	113	4	0.0036	*DSC1|SPRR1B|DSP|JUP*
GO:2001235	Positive regulation of apoptotic signaling pathway	180	4	0.0162	*RPS3|CLU|GSN|YWHAZ*
GO:0071345	Cellular response to cytokine stimulus	1013	7	0.0289	*CANX|RPS3|HNRNPA2B1|ARG1|GSN|YWHAZ|CFL1*
GO:0030155	Regulation of cell adhesion	712	6	0.0328	*IQGAP1|RPS3|ARG1|TACSTD2|GSN|JUP*
**Reactome pathways**					
HSA-168256	Immune System	1956	18	4.76 × 10^−11^	*BPIFB2|CANX|TUBA4A|BPIFA2|DSC1|IQGAP1|HK3|CD14|CLU|HNRNPA2B1|ARG1|HRNR|GSN|DSP|JUP|YWHAZ|CFL1|SERPINB6*
HSA-6809371	Formation of the cornified envelope	127	4	0.0083	*DSC1|SPRR1B|DSP|JUP*
HSA-109581	Apoptosis	173	4	0.0159	*CD14|GSN|DSP|YWHAZ*
HSA-447115	Interleukin-12 family signaling	56	3	0.0159	*CANX|HNRNPA2B1|CFL1*
HSA-6803157	Antimicrobial peptides	87	3	0.0338	*BPIFB2|BPIFA2|CLU*
HSA-76002	Platelet activation, signaling and aggregation	260	4	0.0461	*TUBA4A|CLU|YWHAZ|CFL1*

FDR corrected with Benjamini–Hochberg. Term size: the total number of proteins in the GO and reactome pathways; Enriched terms: number of terms pertaining to the backbone network; Associated genes: proteins associated to the specific term.

## Data Availability

The mass spectrometry proteomics data have been deposited to the ProteomeXchange Consortium via the PRIDE [72] partner repository with the dataset identifier PXD039847.

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
