# Peer review of "Combined Salivary Proteome Profiling and Machine Learning Analysis Provides Insight into Molecular Signature for Autoimmune Liver Diseases Classification"

_ijms, 2023, doi:10.3390/ijms241512207_

Round 1
Reviewer 1 Report
Authors have presented the result from acid-insoluble salivary proteomic profiling to differentiate between patients with primary biliary cholangitis (PBC), autoimmune hepatitis (AIH), and healthy controls (HCs). Authors reports finding a panel of proteins related to immune regulation etc associate able to differentiate patients with PBC from AIH and HCs. For research work with such quality and rigor, the presentation of the manuscript does not provide matching level of attention. Further specific comments are below.
MAJOR ISSUES
1. Overall, the manuscript requires intensive English language proofing and rephrasing to improve clarity. Some specifics are provided below.
2. Section 2.5: in the functional and pathway enrichment analysis, it'd be interesting to understand which proteins are the more central or crucial in the network in terms of topology measures like in-between centrality etc. This may provide details of crucial proteins that may be of pharmaceutical relevance etc (see ttps://doi.org/10.1186/s12859-019-3146-1).
In summary, did the authors consider topology-based functional enrichment analysis and if not, why?
3. Lines 321-322: The innate immune system and neutrophils are part of the immune system's function. Please review statement to be more specific in all the manuscript.
4. Table 4: Please provide the details of the abbreviations under the table for clarity.
5. Please correct all manuscript regarding the term "acidic-insoluble", did you mean to say "acid-insoluble" or "acid-soluble"?
6. Lines 342-351: As opposed to simply repeating the statement already in the result section, authors could rephrase this paragraph to give the general interpretations of the findings in real physiological sense e.g., give the general overview of the combine functionality of the enriched proteins or pathways such as maybe related to immune regulation or response etc.
7. Lines 356-365 falls under the "highly unnecessary information" class. Instead of explaining the mathematical basis and limitations of the method of analysis, authors could devout more time in contextualizing the findings. Readers can sort out more information on the RF if they so wish.
8. Lines 366-371 is simply a repetition of the result section.
9. Lines 375-401: Describing the BPIFA2 and BPIFB2 protein class and function could be summarized into few lined paragraphs and then followed by line 402.
This should be like description of CD14 in lines 417-425.
Essentially, for a PhD thesis most of this information may be relevant but for a journal paper, the readership is different and focusing concisely on the contribution of your work is crucial for readership and future use/references.
10. Lines 415-427: Authors could discuss the relevance of the increase CD14 in PBC with focus on previous studies which have shown similar findings (e.g., DOI: 10.1111/liv.13316).
11. Lines 429-440: This could be summarized again to focus on the context in AIH.
12. Lines 459-461: There is no need to describe the structure here as this is not a protein structure characterization study (also the statement seemed slightly modified from doi:10.1186/s12885-018-4719-5).
This is a clinically relevant study with the prospect of helping refine diagnosis and maybe even prognosis of patients with PBC and AIH.
Overall focus should be more on findings, pathophysiological insight/context etc.
13. Lines 468-469: Again because of the lack of focus, author have discussed here an increase expression of HRNR in cancers while the study show reduction in AIH and PBC. This is a quite interesting contradiction and possibly points to a fundamental difference in the pathophysiological pathways of cancer and PBC/AIH.
Again, I would focus on the disease in which the HRNR have been reduced and the implication etc.
14. Line 479: If no evidence exists regarding the assumption then don't make it. or at least explain why you assume so thoroughly and logically.
15. Line 596: "its"? I supposed these are human subjects.
16. Overall, the manuscript while providing relevant details defeats the purpose of information transfer. The authors have flooded the manuscript with both relevant and totally unnecessary information making readership quite strenuous and poor overall.
I strongly recommend authors to summarize most part and very much avoid repetitions.
This might resolve the problem with too many abbreviations (over 20 of those) which is hard to keep up with.
Also, because of the high abbreviation count, authors could consider following the classic, simple format of allowing method to come before result. Again, there are details in the methods that could be skipped or provided as supplement to avoid information overload.
MINOR ISSUES
i. Line 287: "...composes nodes indicated..." or rephrase.
ii. Line 288: Spell out "PPI".
iii. Lines 294-295: Innate immune system is a type of immune system. Please update/refine this statement.
iv. Line 301: Please correct this sentence "retrieved resulting that".
v. Fig. 5: Please arrange the bar charts in orderly manner.
vi. Lines 317-318: This statement is unnecessary "with the proteins selected by 317 Boruta algorithm as the most discriminating the groups (17 proteins, fig. 1),...".
vii. Lines 322-323: Please clarify the meaning of this statement "belong to these pathways 21 proteins out 23 equally distributed between".
viii. Line 354: "...with good accuracy...".
ix. Lines 444-445: Please clarify this statement which seems incomplete.
x. Line 478: "hypnotized"?
xi. Lines 503-511: How does this relate to PBC and AIH progression?
xii. Line 527: "severs"?
xiii. Lines 546-547: Please clarify this statement "more recently has been evidenced that functional loss of the Trop-2 gene leads to decreased expression...".
xiv. Lines 550: Please provide more references to justify the term "growing body of evidence".
xv. Line 559: ...PBC with 95%...
xvi. Line 593: The statement "Were included in the study patients showing..." is better as "Included in the study are patients that showed...".
xvii. Lines 607: 9:00 am to 1:00 pm or 9:00 to 13:00?
xviii. Line 619: spell out TPC.
General proofing is required.
Author Response
"Please see the attachment."

Reviewer 2 Report
The manuscript titled “Combined salivary Proteome profiling and Machine Learning analysis provides insight into molecular signature for Autoimmune Liver Diseases classification” by Guadalupi et al., investigated the quantitative variations of the salivary proteome for two diseases such as autoimmune hepatitis (AIH) and primary biliary cholangitis (PBC). The manuscript is well written; however, authors need to address a few points before it needs to get accepted.
· It’s better that authors need to show in graphical representation instead of Tabular format for results in Table 2.
· Typo “pairwise” instead of “parwise” for Table 2 caption.
· Figure 1. (a) and (b) are mentioned in the Figure 1 caption, however, authors missed to add in figure.
· Figure 2. X-axis and Y-axis labels were missing.
· Authors pay attention to typos.
Manuscript is well written, however, authors need to pay attention for typos.
Reviewer 3 Report
The authors have investigated the salivary proteome in patients with autoimmune hepatitis (AIH) and primary biliary cholangitis (PBC) compared to healthy controls (HCs). They aimed to identify quantitative variations in salivary proteins associated with these liver diseases and assess the feasibility of using salivary proteins to discriminate between the groups. Overall, the study is impressive but following points need to be critically addressed:
1. What criteria were used to obtain the protein-protein interaction (PPI) networks in the study? How were the interactions determined based on experimental, co-expression, and co-occurrence evidence?
2. How were the clusters in the PPI network automatically subdivided, and what criteria were used to assign different node colors to each cluster? Additionally, what were the total numbers of nodes and edges in each cluster?
3. According to the Reactome pathways enrichment analysis, which biological pathways showed the strongest statistically significant enrichment and the highest number of associated proteins in the red, green, and blue clusters? What are the specific functions or processes related to these pathways?
4. How were the relative importance scores of the selected proteins for classification evaluated in the Random Forest (RF) analysis? What parameters were used to determine the importance of a protein in the model?
5. In Figure 1(b), which specific proteins were identified as the most important for the classification of AIHp and PBCp? How were these proteins determined, and what characteristics made them good discriminant components?
6. What was the classification accuracy, measured by the out-of-bag (OOB) classification error, for AIHp, PBCp, and the combined group of patients versus healthy controls (HCs)? Were there any misclassifications, and if so, what were the error rates?
7. How was the diagnostic value of Hornerin as a potential salivary biomarker for autoimmune liver diseases assessed? What was the area under the ROC curve (AUC) for Hornerin, and what were the sensitivity and specificity values at the best cut-off point?
Round 2
Reviewer 1 Report
I congratulate the authors for this brilliant work.
Hopefully the findings can stimulate further discussion and research on the subject.
Best wishes
Reviewer 3 Report
The authors have addressed all the comments and the current version of the manuscript has sufficient merit to be considered for publication in this journal.